# Partial Discharge Wideband Full-Band High-Gain Resonant Cavity UHF Sensor Research

**DOI:** 10.3390/s23156847

**Published:** 2023-08-01

**Authors:** Chengqiang Liao, Lei Zhang, Guozhi Zhang, Changyue Lu, Xiaoxing Zhang

**Affiliations:** 1Hubei Engineering Research Center for Safety Monitoring of New Energy and Power Grid Equipment, Hubei University of Technology, Wuhan 430068, China; liaochengqiang2001@163.com (C.L.); 102100219@hbut.edu.cn (C.L.); xiaoxing.zhang@outlook.com (X.Z.); 2Electric Power Research Institute, Guangxi Power Grid Co., Ltd., Nanning 530023, China; zhang-9136@163.com

**Keywords:** partial discharge, ultra-high frequency, wide bandwidth, full-band high-gain, Fabry–Perot resonant cavity antenna

## Abstract

To meet the real demand for broadband full-band high-gain antenna sensors in the process of partial discharge (PD) Ultra-High frequency (UHF) detection test and online monitoring of power equipment, this paper builds a resonant cavity monopole UHF antenna sensor based on Fabry–Perot resonant cavity antenna technology, conducts the sensor Voltage Standing Wave Ratio (VSWR) optimization study using curved flow technology, conducts the sensor gain optimization study using slot dual resonant structure, and, finally, tests the sensor performance using the built PD detection test platform. The resonant cavity monopole antenna exhibits outstanding VSWR performance in the frequency range of 0.37 GHz–3 GHz, according to simulation and test data: the average gain in the frequency range of 0.3 GHz–3 GHz is 4.92 dBi, and the highest gain at the primary resonant frequency of 1.0 GHz is 7.16 dBi, with good radiation performance over the whole frequency spectrum. The electromagnetic pulse signal sensed by the UHF sensor developed in this paper can demonstrate the energy spectrum distribution characteristics of PD radiation electromagnetic wave signal more comprehensively, laying a firm technical foundation for thoroughly understanding the electromagnetic wave radiation characteristics of various types of PD insulation defects of various power equipment and the selection of a specific direction for its supporting optimization.

## 1. Introduction

Partial Discharge (PD) is one of the main causes of electrical power failures, which can lead to insulation deterioration, electrical faults, and equipment failures, and can even trigger safety risks such as fire and explosion [1,2,3,4].

The Ultra High Frequency (UHF) method is a method to determine the location, nature, and intensity of PD by using UHF sensors to receive high-frequency electromagnetic waves (300 MHz~3 GHz) generated by the PD inside the power equipment. By transmitting the received signals to the detection system they can be analyzed and processed, which has the advantages of high sensitivity and strong anti-jamming ability, and has been widely used in the power field and laboratory PD detection test has been widely used [5,6,7,8].

However, the area where PD is generated by power equipment is not fixed, resulting in the energy spectrum distribution of the electromagnetic wave signals radiated by PD in different types and different insulation environments also not being fixed. Although the International Electrotechnical Commission standard IEC 60270-2015 and China’s national standard GB/T 7354-2018 both put forward explicit requirements for the sensitivity of UHF sensors in the 300 MHz–3 GHz band range [9,10], from the long-term field practice, the current UHF sensors that meet the requirements of the standard are unable to meet the demand for high sensitivity.

More importantly, although a large number of currently developed UHF sensors meet the requirements of the relevant standards, the volatility between the corresponding sensor performance parameters (VSWR, gain, etc.) at different frequency points or different frequency bands means that the PD high-frequency electromagnetic wave signals obtained from power field PD monitoring or laboratory PD detection cannot fully respond to the PD radiation electromagnetic wave energy spectrum characteristics, and cannot guide the subsequent optimization of UHF sensors for different types of PD detection in diverse power equipment (for example, the back-cavity antenna, which is often used to increase the gain of sensors, is only effective in some frequency bands) [11]; therefore, it is very necessary to research wideband full-band high-gain UHF sensors [12,13].

To address the above problems, this paper carries out research on the design of broadband high-gain monopole UHF antenna based on the Fabry–Perot resonant cavity antenna design idea, carries out sensor VSWR optimization using curvilinear flow technique and sensor gain optimization using slot dual resonant structure, and uses High Frequency Structure Simulator (HFSS) to optimize and simulate the antenna structure, in the end, the designed antenna is prototyped and the performance of the antenna is tested and analyzed using a vector network analyzer and PD simulation experimental platform.

## 2. Principle of Antenna

### 2.1. The Basic Principle of Monopole Patch Antenna

A monopole patch antenna is a very common antenna, which usually consists of a radiating patch on a thin dielectric substrate and a relatively grounded metal floor for receiving electromagnetic signals propagating in space and feeding through coaxial probes, microstrip, or coplanar waveguides, with the advantages of a low profile, small size, and simple fabrication [14].

The planar monopole patch antenna can be approximated with an equal height cylindrical structure [15], as shown in Figure 1, where the orange area in Figure 1b is the metal radiating electrode and the yellow is the metal floor. When the cylindrical structure is cut and expanded along its height, the rectangle obtained from this cylindrical structure is the estimated rectangular radial patch:(1)2πRL=ab
where *R* is the base radius of the equivalent cylinder, *L* is the height of the equivalent cylinder, *a* is the width of the radiating patch of the monopole antenna, and b is the length of the radiating patch.

The real part of the input impedance of the monopole patch antenna is slightly less than 1/4 wavelength monopole length while considering the gap *g* between the metal floor and the radiation patch, the lowest working frequency *f_L_* of the antenna is obtained as follows:(2)$$f_{L}=\frac{c}{\lambda }=\frac{72}{L+R+g}$$
where *c* is the speed of light and *λ* is the wavelength corresponding to the lowest operating frequency. After considering the effect of the dielectric substrate on the resonant frequency of the antenna, Equation (2) is corrected, that is
(3)$$f_{L}=\frac{c}{\lambda }=\frac{72}{\left(L+R+g\right)\sqrt{\varepsilon_{eff}}}$$
where *ε_eff_* is the equivalent dielectric constant. It can be estimated as
(4)$$\varepsilon_{eff}=\frac{\varepsilon_{r}+1}{2}$$
where *ε_r_* is the relative permittivity of the substrate.

The above theory can be used to estimate the starting size of the monopole patch antenna. Subsequently, the design can be optimized on this basis to achieve the desired performance.

### 2.2. The Basic Principle of Microstrip Feeder Transmission Line

The microstrip feed line is a common way to feed antennas, its basic principle is to achieve the feed by laying metal strip lines on the dielectric substrate. When the electromagnetic wave propagates to the microstrip feed line, a surface wave is generated between the microstrip line and the ground plane, and this surface wave propagates along the microstrip line and eventually reaches the antenna radiation element. The microstrip transmission line width *w* and the microstrip transmission line characteristic impedance *z* satisfies:(5)$$z=\frac{120\pi }{\sqrt{\varepsilon_{eff}}\left[\frac{w}{h}+2.42-0.44\frac{h}{w}+{\left(1-\frac{h}{w}\right)}^{6}\right]}$$
where *h* is the thickness of the dielectric substrate. The desired impedance, dielectric substrate thickness, and equivalent dielectric constant are brought into the equation to calculate the microstrip line width for good impedance matching.

### 2.3. The Basic Principle of Fabry–Perot Resonant Cavity Antenna

The Fabry–Perot resonant cavity is a multi-beam optical interferometer that can be used to accurately measure the wavelength and refractive index of light, invented by French physicists Charles Fabry and Alfred Perot in 1897 [16]. The interferometer consists of two parallel mirrors and the space between them forms a resonant cavity. Under certain conditions, the light rays in the resonant cavity can form a series of beams that interfere with each other [17]. This principle can be used in optical sensors to improve filtering performance and increase gain [18].

Later, research scholars discovered that a similar principle could be applied to the field of antenna sensors, leading to the invention of the Fabry–Perot resonant cavity antenna. In the literature [19], Li Lu loaded a Fabry–Perot resonant cavity on an LTCC microstrip antenna [19], and the resulting resonant cavity gathers the radiated energy inside the cavity to realize the radiation and reception of the signal at the resonant frequency, which improved the strength of the electromagnetic field and achieved the advantage of high gain at low profile. In the literature [20], Yanru Hua designed a partially reflective surface unit structure [20] showing a rising phase, which is loaded above the feed antenna to form a Fabry–Perot resonant cavity antenna, overcoming the narrow band of the conventional Fabry–Perot resonant cavity antenna. The principle of the Fabry–Perot resonant cavity antenna is described in detail below:

The Fabry–Perot resonant cavity configuration is shown in Figure 2 and consists of a partially reflective cladding, a feed antenna and its floor. Part of the reflective cladding is parallel to the feed antenna at a distance of *l,* between which multiple reflections of electromagnetic waves occur. The feed antenna and its floor are a fully reflective plate with a reflection coefficient of $e^{j\varphi_{1}}$, the amplitude of the reflection coefficient is 1, and the phase is $\varphi_{1}$. The partially reflective cladding is a partially reflective plate with a reflection coefficient of $pe^{j\varphi_{2}}$, the amplitude of the reflection coefficient is *p*, and the phase is $\varphi_{2}$. Assuming no transmission loss of the electromagnetic wave, the amplitude of the transmitted beam 0 is proportional to $\sqrt{1-p^{2}}$; after one reflection, the amplitude of the transmitted beam 1 is proportional to $p\sqrt{1-p^{2}}$; after two reflections the amplitude of the transmitted beam 2 is proportional to $p^{2}\sqrt{1-p^{2}}$.

After *n* reflections, the superposition of the electric field intensity of all transmitted waves transmitted through the partially reflected layer gives:(6)$$E=\sum_{n=0}^{\infty }F\left(\alpha \right)E_{0}p^{n}\sqrt{1-p^{2}}e^{j\theta_{n}}$$
where *F*(*α*) is the feed antenna radiation direction function, *E*_0_ is the maximum radiation field strength, *θ_n_* is the phase difference between the transmitted wave beam n, and the transmitted wave beam 0
(7)$$\theta_{n}=n\varphi =n\left[\frac{4\pi }{\lambda }l\cos\alpha -\varphi_{1}-\varphi_{2}\right]$$
as *p* < 1, we can obtain:(8)$${\sum_{n=0}^{\infty }\left(pe^{j\varphi }\right)}^{n}=\frac{1}{1-pe^{j\varphi }}$$
substituting Equation (8) into Equation (6) yields the modal value of the electric field:(9)$$\left|E\right|=\left|E_{0}\right|f\left(\alpha \right)\sqrt{\frac{1-p^{2}}{1+p^{2}-2p\cos\varphi }}$$
substituting Equation (7) into Equation (9) yields the energy density in the *α* direction:(10)$$S=\frac{1-p^{2}}{1+p^{2}-2p\cos\left(-\varphi_{1}-\varphi_{2}+l\frac{4\pi }{\lambda }\cos\alpha \right)}f^{2}\left(a\right)$$
usually, the maximum radiation direction of the Fabry–Perot resonant cavity antenna is *α* = 0°, and *S* is maximum when $\varphi_{1}+\varphi_{2}-\left(4\pi /\lambda \right)l=2n\pi$, *n* = 0, 1, 2, … are satisfied, and its maximum value is
(11)$$S_{\max}=\frac{1+p}{1-p}f^{2}\left(0\right)$$
It can be seen that the transmitted power coefficient of the resonant cavity antenna is strongly related to the reflection coefficient amplitude *p* of the partially reflected surface [21]. The greater the amplitude of the reflection coefficient of the partially reflective cladding, the greater the transmission power coefficient of the resonant cavity antenna, and the higher the gain. Therefore, it is necessary to improve the gain of the antenna through the reasonable design of part of the reflective cladding so that the transmitted wave of the feed antenna through the upper part of the reflective plate is continuously transmitted and superimposed in the same phase.

## 3. Resonant Cavity UHF Antenna Sensor Design

The resonant cavity UHF antenna sensor designed in this paper is a Fabry–Perot resonant cavity monopole antenna sensor, which contains two parts: the feed antenna and the partial reflection cladding. The feed antenna is a microstrip monopole patch antenna with optimized VSWR using curved flow technique and optimized gain using slot dual resonant structure.

### 3.1. Microstrip Monopole Patch Antenna Body Design

#### 3.1.1. Antenna Design Based on Curved Flow Technology

The dielectric substrate of the microstrip monopole antenna designed in this paper is rectangular rigid material FR-4 epoxy resin, the relative dielectric constant of the substrate *ε_r_
*= 4.4, the dielectric loss tangent angle tan*δ* = 0.03, and the thickness of the dielectric substrate *h* = 2.8 mm. The microstrip monopole antenna designed in this paper uses SMA coaxial feed with a characteristic impedance of 50 Ω.

Since the energy of the PD signal is concentrated in the range of 500 MHz–1.5 GHz, and the common monopole patch antenna is often used in the frequency band above 3 GHz in the communication field, if we want to make it detect partial discharge, we need to extend its working band.

In this paper, the patch of monopole patch antenna in Figure 1 is slotted, and a stepped structure is added by using the curved flow technique, and finally the antenna is constructed by using HFSS software as shown in Figure 3.

Curved flow technology is a technique to make the surface current flow along a specific structure on the metal patch of microstrip antenna by making slots or adding slits, introducing multi-step structure, etc. Such a design can free the equivalent length of the antenna from the physical length limitation, thus effectively increasing the equivalent length of the microstrip antenna and improving the bandwidth of the antenna without increasing its size [15,22,23,24]. Each parameter of the monopole patch antenna based on the curved flow technique is shown in Table 1.

HFSS software is used to simulate and analyze the antenna with the above parameters and structure, and the data of VSWR is shown in Figure 4. The VSWR of the monopole antenna improved by the curved flow technology is less than 5 in 380 MHz–3 GHz, and the working band of the antenna basically covers the frequency band of the UHF signal, which meets the design requirements of the PD UHF sensor for power equipment and can be optimized subsequently.

Figure 5 shows the simulated gain direction of the antenna at the center frequency point of 1 GHz, the highest gain is only 1.27 dBi, and the directionality is poor, necessitating subsequent optimization to improve the directionality of the antenna gain.

#### 3.1.2. Structural Optimization Based on Slot Dual Resonance

The traditional slot double resonant structure is in the antenna surface of an artificially created rectangular groove: the groove will form a resonant cavity, so that the propagation of electromagnetic wave resonance thus increasing the antenna gain, but this practice will make the antenna bandwidth greatly reduced. Therefore, this paper uses the idea of slot dual resonant structure to form a slot dual resonant structure similar to the effect of a concave resonant cavity by adding a circle of peripheral patches around the microstrip monopole antenna as shown in Figure 6, while setting isosceles triangle structures on the top left and top right of the added peripheral patches and rectangular arm structures on the left and right sides [16,25,26,27]. The improved slot dual resonance structure can concentrate the slot current to the center patch position and increase the center current density, which can also improve the gain of the antenna sensor.

The geometric parameters of the slotted dual resonant structure of the antenna in Figure 6 are given in Table 2.

The VSWR and gain performance of the improved monopole antenna are simulated and analyzed. From the VSWR simulation plot in Figure 7 below, it can be seen that the VSWR fluctuation becomes larger compared with that before the improvement, but the VSWR is less than 5 in the 330 MHz–3 GHz band, which basically covers the frequency band of UHF signal and meets the design requirements of PD UHF sensor for power equipment. As can be seen from the gain direction diagram in Figure 8 below for the center frequency point of 1 GHz, the maximum gain is increased by 1.81 dBi to 3.08 dBi compared to that before the improvement, and the gain direction diagram is in the shape of “8” with good directionality, which is basically in line with the improvement direction to improve the gain performance.

### 3.2. Partially Reflective Cladding Design and Combination

The general partial reflection cladding with the feed antenna has a narrow bandwidth, which is because the reflection coefficient of the partial reflection layer becomes smaller in phase as the frequency increases and is a negative reflection phase gradient [18]. For this defect, a partially reflective cladding with a positive reflective phase gradient is designed in this paper.

The partially reflective cladding unit designed in this paper adopts a single-layer double-sided asymmetric square structure, while empirically loading the perforations to make the gain performance smoother, and its structure is shown in Figure 9.

To meet the desired direction of improvement, the front square patch width *a* is 60 mm and the back square patch width *b* is 64 mm. The distance of the perforation with a diameter of 2 mm from both sides of the square patch is 20 mm in both *x* and *y*.

The front unit structure is expanded into a uniformly distributed 3 × 3 partial reflection cladding. The selection of dielectric substrate material is the same as the microstrip monopole antenna, i.e., FR-4 epoxy resin. The dimensions of the whole partial reflection cladding are also consistent with the microstrip monopole antenna, length × width × thickness = 205.2 mm × 241.2 mm × 2.8 mm, and the structure of the composed partial reflection cladding is shown in Figure 10, where *W*_b_ = 20.6 mm and *L*_b_ = 2.6 mm.

The feed antenna designed in Section 3.1 and the partial reflection cladding designed in Section 3.2 are combined together by nylon screws, and the height between the lower feed antenna and the upper partial reflection cladding is d = 25 mm, which constitutes the physical structure of Fabry–Perot resonant cavity monopole antenna, as shown in Figure 11 below.

## 4. Antenna Performance Simulation Analysis

### 4.1. Voltage Standing Wave Ratio

Voltage Standing Wave Ratio (VSWR) is a parameter used to describe the ratio of reflected signal strength and forward signal strength at the input port of the antenna, measuring the impedance matching ability between the input and output circuits of the antenna. When the signal enters the transmission line and reaches the antenna sensor, part of the signal will be reflected back to the transmission line, while the other part will be transmitted into the air; these reflected waves will be superimposed with the original waveform, creating the so-called “standing wave”, which has a negative impact on the transmitted energy and signal. VSWR can be calculated by the following equation:(12)$$\mathrm{VSWR}=\frac{{\left|U\right|}_{\max}}{{\left|U\right|}_{\min}}=\frac{1+\left|\Gamma \right|}{1+\left|\Gamma \right|}$$
where Γ is the reflection coefficient of the antenna input, defined as the ratio of reflected power to incident power. The lower the voltage VSWR, i.e., the closer to 1, the better the impedance matching effect of the input port, that is, more electromagnetic waves are transmitted to the antenna. On the contrary, the higher the voltage VSWR, the more impedance mismatch, indicating that more signals are reflected back, which will degrade the antenna performance, leading to signal loss, increased bit error rate, and other problems, and may even cause damage to the circuit.

In addition, since the UHF antenna sensor operates in a UHF environment, the signal attenuates quickly, so it is necessary to ensure that as much signal as possible is transmitted to the antenna to ensure the sensitivity and accuracy of the antenna. Therefore, a lower VSWR value is critical for the quality and stability of the transmission circuit in this case, so a lower VSWR is a fundamental requirement for the design and use of high-performance UHF antenna sensors. When VSWR ≤ 2 the antenna sensor receives the best electromagnetic energy, but in fact VSWR ≤ 5 can meet the engineering needs, so this paper will use VSWR ≤ 5 as the PD detection bandwidth of power equipment [28,29,30,31,32,33,34].

The Fabry–Perot resonant cavity monopole antenna model in Section 3 above is constructed using HFSS simulation software, and the VSWR curves are obtained from its simulation in the frequency range of 300 MHz~3 GHz as shown in Figure 12. The simulation results show that the VSWR of the UHF antenna designed in this paper is less than 5 in the frequency range of 0.41 GHz–1.69 GHz and 1.75 GHz–3 GHz, which meets the design requirements.

The VSWR of the physical antenna was measured using an E5063A vector network analyzer manufactured by Agilent with the same sweep range of 300 MHz–3 GHz. As shown in Figure 13, in the 0.37 GHz–3 GHz VSWR ≤ 5, VSWR performance is good. Comparing the measured and simulated results, the overall trend of the measured VSWR results is basically the same as the simulated results, and the measured results are basically better than the simulated ones in the whole measurement band, but the oscillation is more obvious and meets the requirements of engineering design.

### 4.2. Gain Direction Diagram

The gain directional map is a parameter used to describe the directionality of the antenna and the antenna efficiency, measuring the degree of concentration of the radiant energy of the antenna in free space and the degree of effectiveness of the antenna in converting wave energy into high-frequency current or energy [18,35,36]. The higher the gain, the better the directionality, the more concentrated the energy, and the higher the efficiency. At the same time, the higher the gain means the higher the receiving sensitivity of the antenna, and the more favorable to the processing of PD signal. Therefore, the antenna has high gain performance is the common requirement of receiving antenna and transmitting antenna. There are already many methods to evaluate antenna radiation, such as near-field to far-field transformation techniques [37,38].

The gain direction plots of the designed Fabry–Perot resonant cavity monopole antenna at six frequency points of 0.7 GHz, 1.0 GHz, 1.4 GHz, 2.0 GHz, 2.4 GHz, and 3.0 GHz are shown in Figure 14. As can be seen from the figure, the antenna gain direction shows symmetrical characteristics, 0.7 GHz and 1.0 GHz in the low-frequency band have the maximum gain at 0° in the axial direction; the antenna is inverted “8” at 1.0 GHz, with good bi-directional radiation characteristics. The gain direction map starts to show fission and depression at 1.4 GHz, which may be caused by too much cross-polarization due to the complexity of the current path.

The HFSS simulation shows that the average gain of the designed resonant cavity monopole antenna is 4.92 dBi in the frequency band of 0.3 GHz–3 GHz, and the maximum gain at each frequency point obtained from the simulation is given in Table 3. As shown by the simulation results, the peak gain at the main resonant frequency of 1.0 GHz is 7.16 dBi, and the gain is relatively high in the whole frequency band, which has a high gain performance. This advantage allows the Fabry–Perot resonant cavity monopole antenna to detect the PD signal to a larger area in the surrounding area, achieving higher sensitivity and better signal-to-noise ratio.

## 5. PD Detection Performance Test

### 5.1. PD Inspection Test Platform

To verify the performance of the designed Fabry–Perot resonant cavity monopole antenna for detecting PD UHF signals, an industrial frequency, and high voltage testbed without local discharge was built in the laboratory, as shown in Figure 15. The experimental circuit consists of an industrial frequency AC power supply, a filter T_1_, a regulator, a step-up transformer T_2_, a protection resistor R_r_, voltage divider capacitors C_1_ and C_2_, and a cavity that simulates a GIS structure. The PD signal from the antenna sensor is fed through a coaxial cable to the signal acquisition device Tektronix*MSO44 (oscilloscope with four collectible channels, sampling frequency of 6.25 GS/s, bandwidth of 1.5 GHz) Tektronix high-performance digital oscilloscope for acquisition. The GIS simulation chamber tank is filled with 0.5 MPa of SF_6_ gas with 99.999% purity. The experiment uses the air gap discharge model to simulate the insulation fault PD process inside the GIS equipment. The air gap defect is caused by the substandard process of GIS tub insulators, which produces air bubbles during the curing of epoxy resin and the air bubbles evolve into air gaps, and PD will occur in these air gaps during the operation of GIS. The air gap defect model used in this test is as follows: there are three pieces of epoxy resin sandwiched between the high-voltage electrode and the ground electrode, and the middle piece has an air gap defect with a volume of 10 mm × 10 mm × 5 mm, which is shown schematically in Figure 16.

The experiment is based on the State Grid Corporation of China’s corporate standard: Q/GDW11304.8-2019 “Ultra-high frequency method of partial discharge Charged detection technical standards” [39].

### 5.2. Antenna Performance Comparison

In this paper, a high-performance flexible monopole antenna designed in the literature [40] is used to perform comparative PD detection performance tests [40]. The two sensors were placed at the viewing window of the GIS-simulated tank of the cavity and then tested. In the voltage range of 3.7 kV–4.3 kV was used to complete 20 PD comparison tests, the data is shown in Figure 17. From the figure, it can be seen that both sensors can effectively detect the PD radiation electromagnetic pulse signal, and that the UHF sensor designed in this paper has a higher sensitivity.

The results of the PD UHF signals detected by both sensors were recorded when the voltage was increased to 3.7 kV and the discharge was about 17.2 pC, as shown in Figure 18. The results show that the maximum signal amplitude measured by the flexible monopole antenna is 14.80 mV, while the maximum signal amplitude measured by the Fabry–Perot resonant cavity monopole antenna is 23.20 mV, both of which decay to the background noise level after about 0.16 μs and 0.30 μs. At the end of the UHF signal of the Fabry–Perot resonant cavity monopole antenna, oscillations were observed, but 20 comparison tests showed that the Fabry–Perot resonant cavity monopole antenna could detect the PD signal and the detection performance of the Fabry–Perot resonant cavity monopole antenna was superior to that of the comparison flexible UHF sensors [41,42].

The spectrum analysis of the detected PD signal and the background noise spectrum analysis are shown in Figure 19. Comparing the pictures, it can be seen that the spectrum distribution of the flexible monopole antenna and the Fabry–Perot resonant cavity monopole antenna are basically the same, and the background noise mainly appears at the two communication interference frequencies of 0.9 GHz and 1.8 GHz. The feasibility of the Fabry–Perot resonant cavity monopole antenna for PD detection and reflecting the electromagnetic wave energy spectrum characteristics of PD radiation is verified.

## 6. Conclusions

For the current demand for UHF antenna sensors for PD detection of power equipment with broadband full-band high-gain antenna sensors, this paper carries out the research of broadband high-gain resonant cavity monopole UHF antenna for PD detection of power equipment, using curved-flow technology and slot double resonant structure to optimize the monopole antenna. Based on the idea of Fabry–Perot resonant cavity antenna for high gain antenna, the designed antenna was analyzed and verified by experiments:Optimization of monopole patch antenna by using the idea of curved-flow technology and slot dual resonance structure, which greatly broadens the operating band of the antenna sensor and improves the antenna gain directionality. Based on the idea of Fabry–Perot resonant cavity antenna, partial reflection cladding was set up, so that the antenna sensor can obtain full-band high gain performance. The simulated and measured results show that the designed resonant cavity monopole antenna has VSWR ≤ 5 in the frequency band of 0.37 GHz–3 GHz, excellent VSWR performance, average gain of 4.92 dBi, peak gain of 7.16 dBi at the main resonant frequency, simple fabrication process, and meets the design requirements of PD UHF antenna sensors.Through the constructed partial discharge test platform, the designed resonant cavity monopole antenna is used to detect the typical air gap discharge defects, and compared with the mature flexible monopole UHF antenna, the results show that the resonant cavity monopole antenna can detect the PD signal more obviously, which proves the feasibility of resonant cavity monopole antenna for PD detection and reflecting the PD radiation electromagnetic wave energy spectrum characteristics.

## Figures and Tables

**Figure 1 sensors-23-06847-f001:**
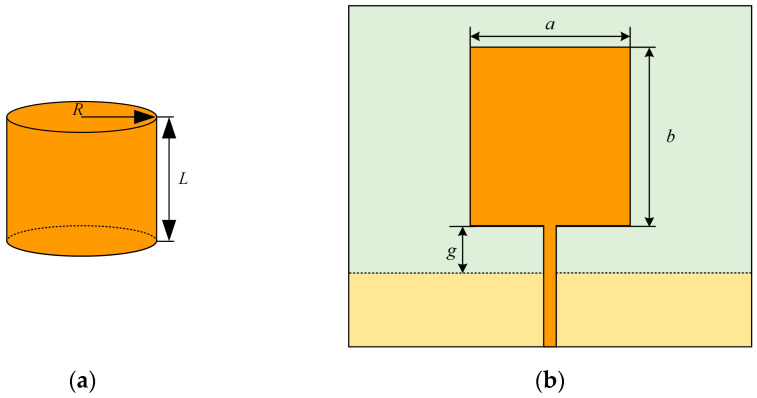
Monopole patch antenna schematic: (**a**) Cylindrical Structure; (**b**) Antenna prototype.

**Figure 2 sensors-23-06847-f002:**
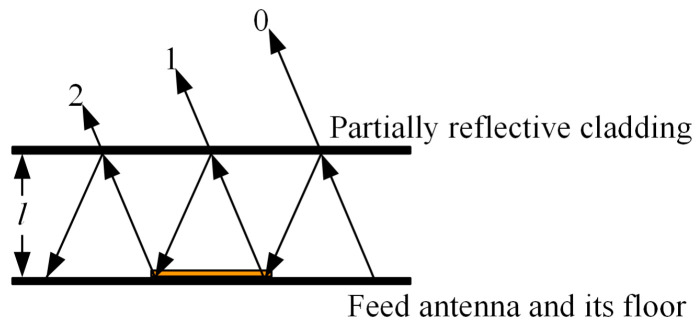
Fabry–Perot resonant cavity antenna structure.

**Figure 3 sensors-23-06847-f003:**
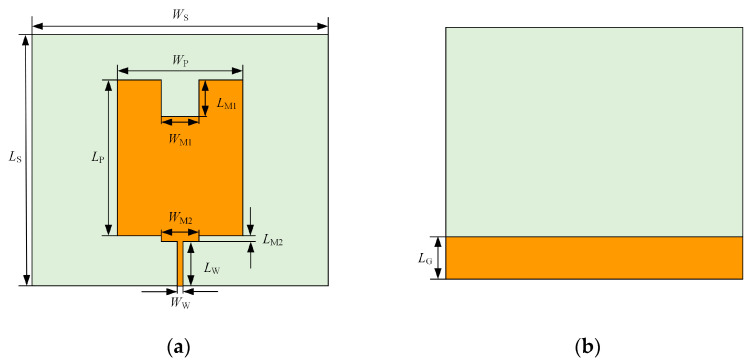
Diagram of monopole antenna based on curved flow technique: (**a**) antenna front; (**b**) antenna back.

**Figure 4 sensors-23-06847-f004:**
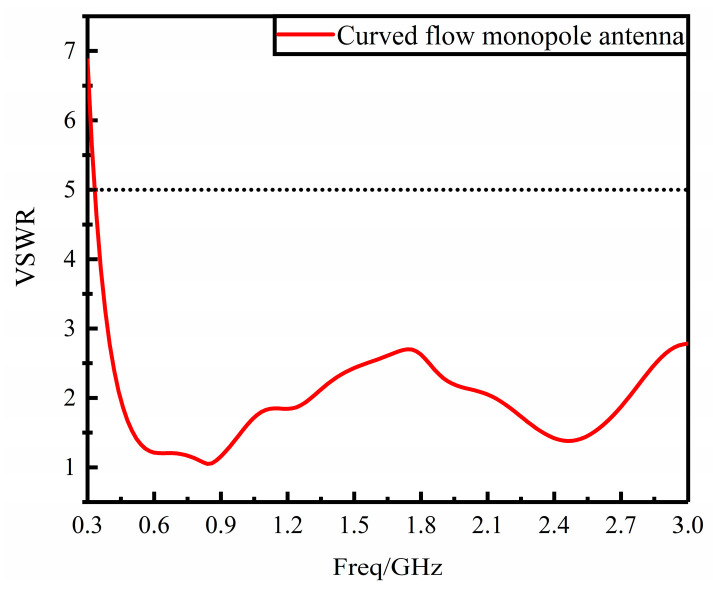
VSWR diagram of curved current monopole antenna.

**Figure 5 sensors-23-06847-f005:**
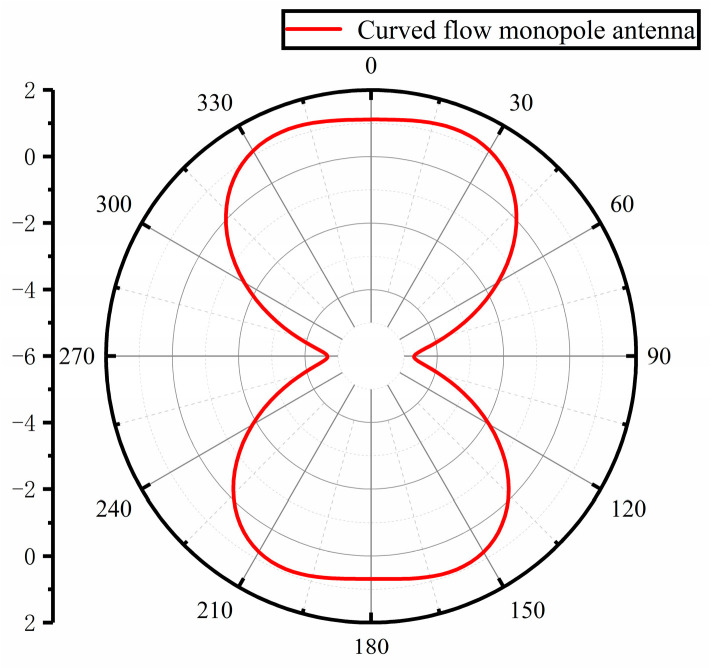
Curved flow monopole antenna gain direction diagram.

**Figure 6 sensors-23-06847-f006:**
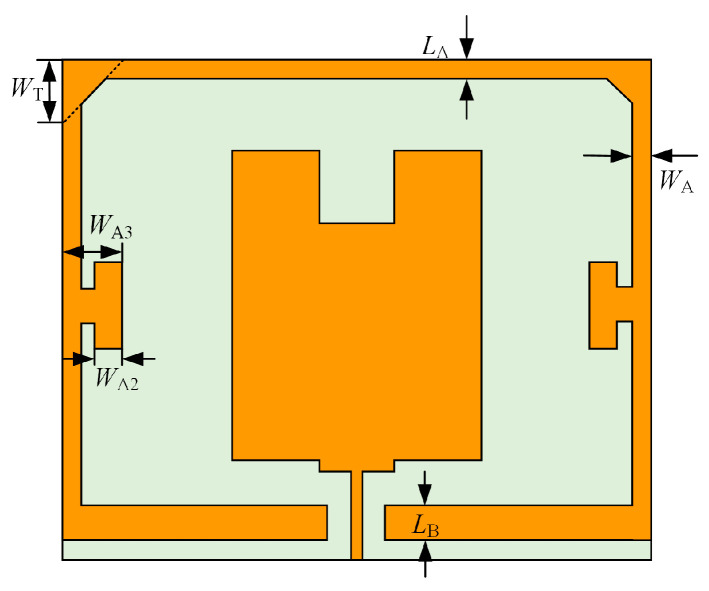
Diagram of monopole antenna based on slot dual resonance structure.

**Figure 7 sensors-23-06847-f007:**
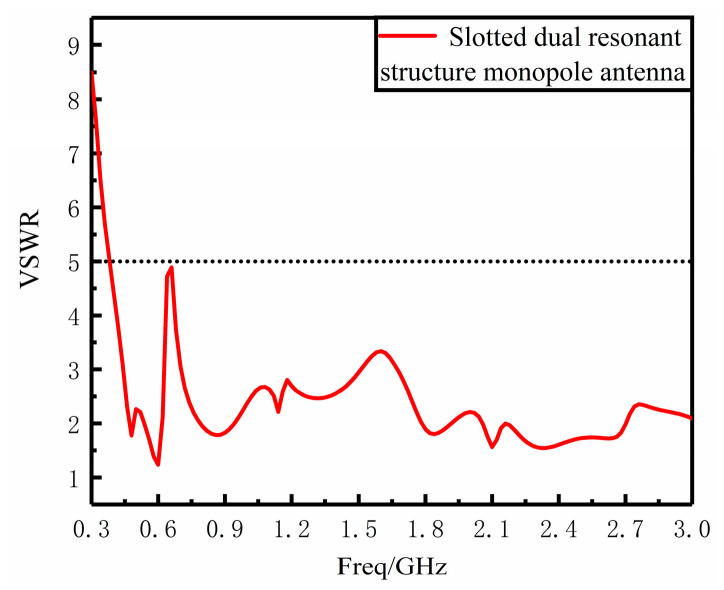
Slotted dual resonant structure monopole antenna VSWR diagram.

**Figure 8 sensors-23-06847-f008:**
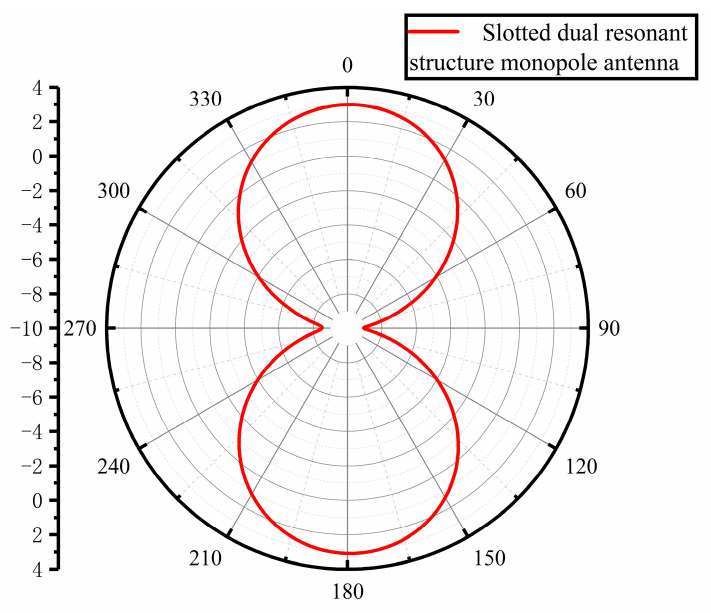
Slotted dual resonant structure monopole antenna gain direction diagram.

**Figure 9 sensors-23-06847-f009:**
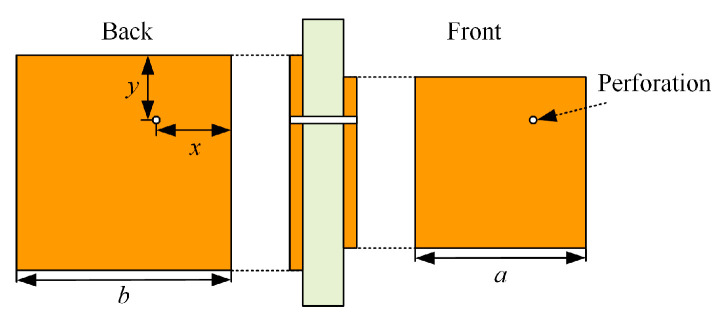
Partial reflective cladding unit diagram.

**Figure 10 sensors-23-06847-f010:**
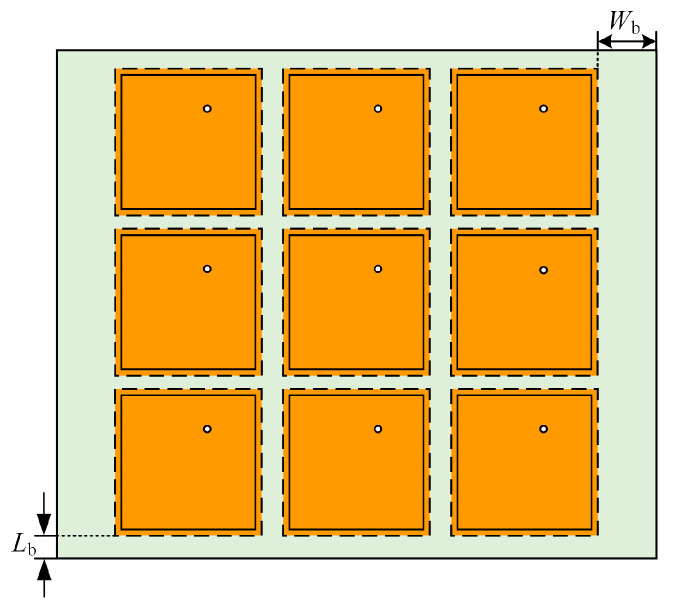
Partial reflective cladding structure diagram.

**Figure 11 sensors-23-06847-f011:**
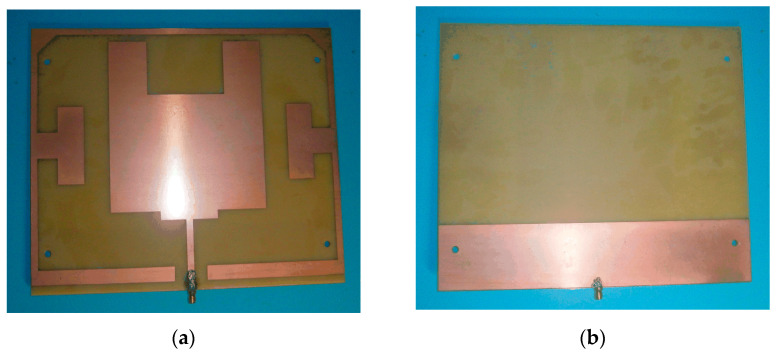
Resonant cavity monopole antenna structure diagram: (**a**) front of the feed antenna; (**b**) back of the feed antenna; (**c**) partially reflective cladding; (**d**) resonant cavity monopole antenna.

**Figure 12 sensors-23-06847-f012:**
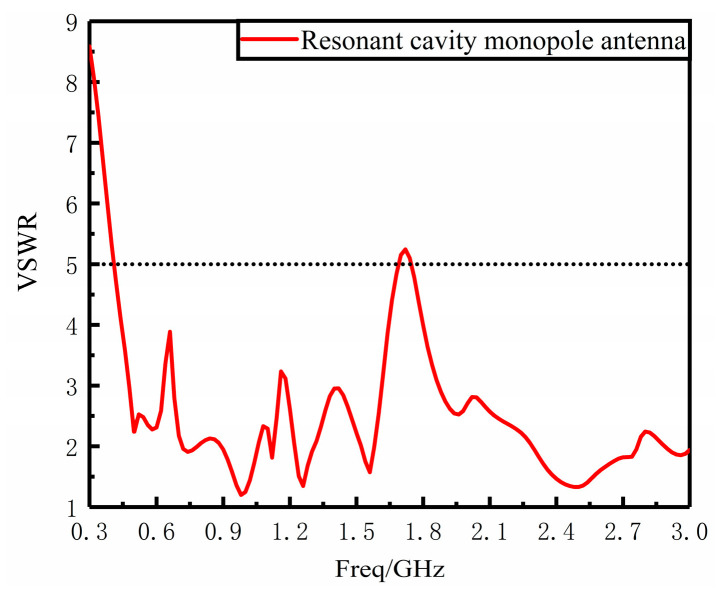
VSWR diagram of resonant cavity monopole antenna.

**Figure 13 sensors-23-06847-f013:**
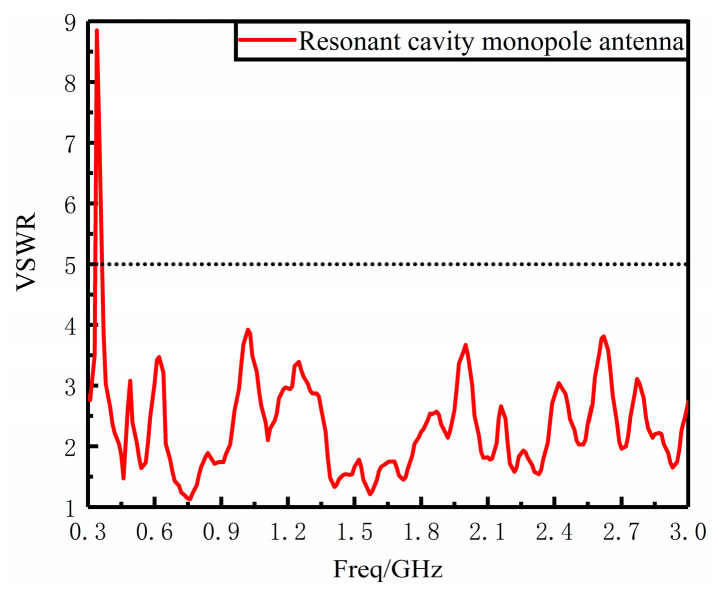
Resonant cavity monopole antenna measured VSWR diagram.

**Figure 14 sensors-23-06847-f014:**
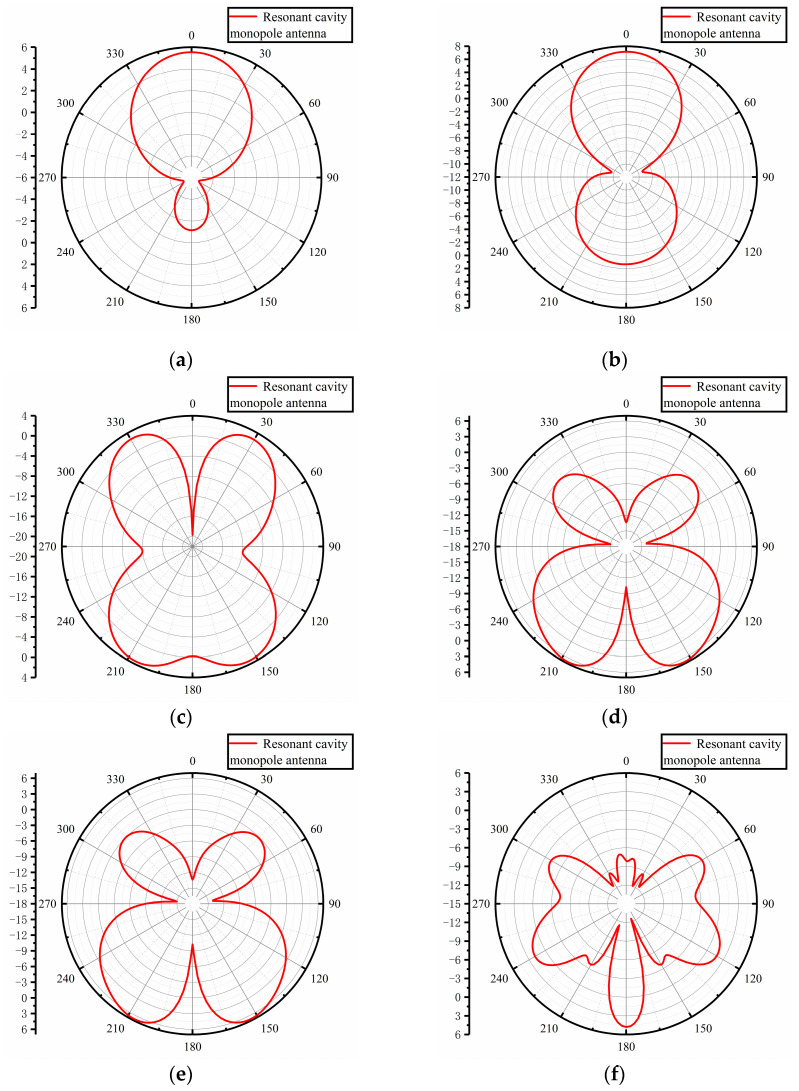
Gain direction diagram at different frequency points: (**a**) 0.7 GHz; (**b**) 1.0 GHz; (**c**) 1.4 GHz; (**d**) 2.0 GHz; (**e**) 2.4 GHz; (**f**) 3.0 GHz.

**Figure 15 sensors-23-06847-f015:**
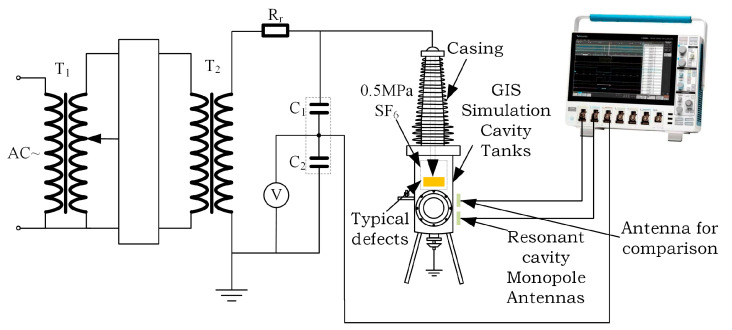
Experimental circuit diagram.

**Figure 16 sensors-23-06847-f016:**
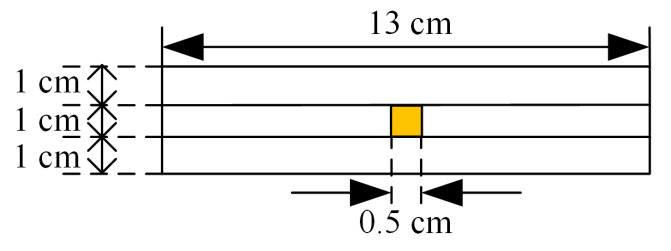
Diagram of air gap defect.

**Figure 17 sensors-23-06847-f017:**
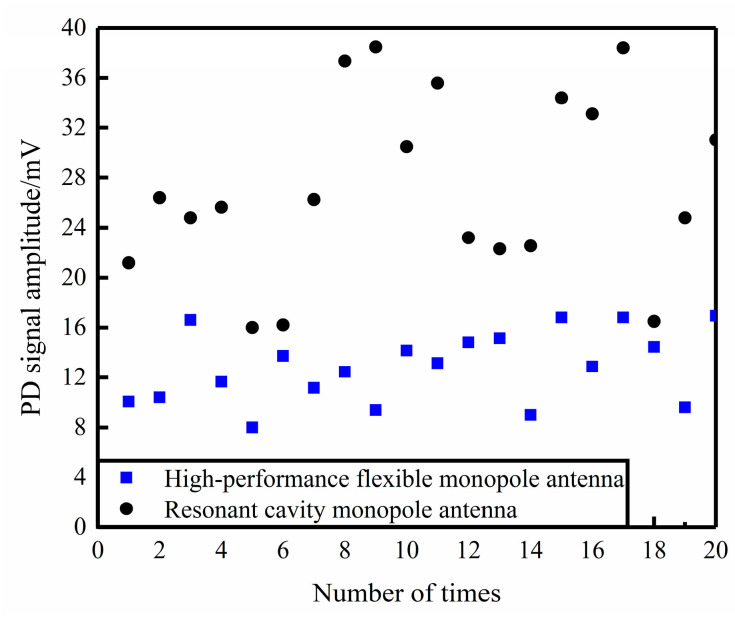
Statistical chart of comparison experiment data.

**Figure 18 sensors-23-06847-f018:**
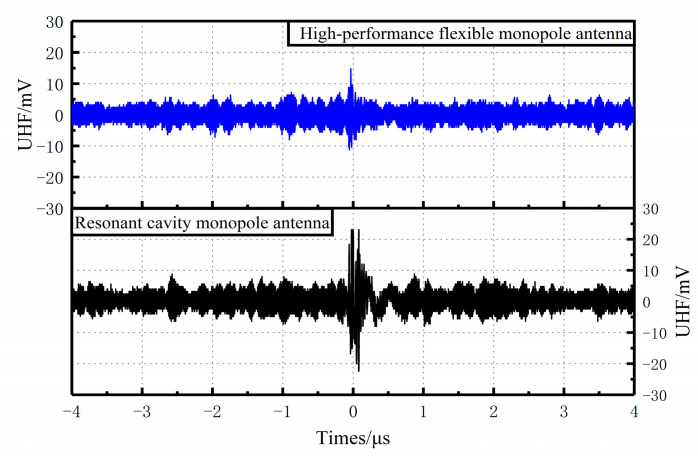
Waveforms of PD signals collected by different sensor antennas.

**Figure 19 sensors-23-06847-f019:**
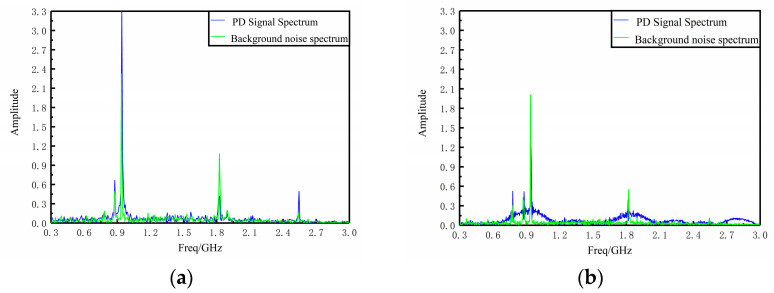
Detection of PD signal spectrum and noise spectrum by different sensor antennas: (**a**) high-performance flexible monopole antenna; (**b**) resonant cavity monopole antenna.

**Table 1 sensors-23-06847-t001:** Geometric parameters of monopole antenna based on curved flow technique.

Parameters	Value (mm)
Substrate length *L*_S_	205.2
Substrate width *W*_S_	241.2
Patch length *L*_P_	134.0
Patch width *W*_P_	120.6
Rectangular slot length *L*_M1_	42.9
Rectangular slot width *W*_M1_	60.9
Step structure length *L*_M2_	6.7
Step structure width *W*_M2_	44.2
Microstrip line length *L*_W_	53.6
Microstrip line width *W*_W_	5.1
Dielectric substrate thickness h	2.8
Floor length *L*_G_	48.9

**Table 2 sensors-23-06847-t002:** Table of geometrical parameters of slot double resonance structure.

Parameters	Value (mm)
Upper border length *L*_A_	5.0
Isosceles triangle side length *W*_T_	25.0
Left and right border width *W*_A_	5.0
Large rectangle arm width *W*_A3_	40.6
Small rectangle arm width *W*_A2_	20.0
Lower border length *L*_B_	11.0

**Table 3 sensors-23-06847-t003:** Maximum gain at different frequency points.

Frequency Points	Maximum Gain
0.7 GHz	5.54 dBi
1.0 GHz	7.16 dBi
1.4 GHz	3.46 dBi
2.0 GHz	6.70 dBi
2.4 GHz	6.71 dBi
3.0 GHz	4.79 dBi

## Data Availability

The data presented in the article is original and has not been inappropriately selected, manipulated, enhanced, or fabricated by us.

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
