# Peer review of "Partial Discharge Wideband Full-Band High-Gain Resonant Cavity UHF Sensor Research"

_sensors, 2023, doi:10.3390/s23156847_

Round 1

Reviewer 1 Report

In this paper, the Authors are building a resonant cavity monopole UHF antenna sensor based on Fabry-Perot resonant cavity antenna technology.

The results of experimental tests show that the proposed method is very promising. Based on the idea of Fabry-Perot resonant cavity antenna to set up partial reflection cladding, so that the antenna sensor can obtain full-band high gain performance.

After carefully reading, I find that this paper is extremely interesting, however in order to further improve I would only recommend to improve the conclusions and more references on the background (I suggest: doi.org/10.3390/s23125361; doi.org/10.1049/iet-map.2016.0936, doi.org/10.3390/s23104722; doi.org/10.3390/s23146493; doi.org/10.1049/iet-map.2015.0825). 

Reviewer 2 Report

Reviewer notes:

In this research a resonant cavity monopole Ultra-High Frequency (UHF) antenna sensor based on Fabry-Perot resonant cavity antenna technology was built, and the sensor VSWR optimization study using curved flow technology has been conducted. At the same time, there are the following comments to improve the quality and presentation of the work:

1)  Title consists of several abbreviations, even first word ?!?

The same about the text, e.g. HFSS.

All abbreviations must be defined!

2)  Abstract:  VSWR.

All abbreviations must be defined!

3) Keywords:  Wideband…

It’s very strange to see one adjective as Keyword…

Remove it or add noun to make the whole phrase clear

PD; UHF -   abbreviations must be written.

4) Section “5. Actual Measurement of Antenna Performance”.

Does it mean that all other measurement NON actual?

5) Nothing is said about UHF method itself.

Please explain more broad than one paragraph in the Introduction, give necessary references.

How it correlates with "Technical Specification for Partial Discharge Strip Testing by UHF 339 Method" [34]?

6) Equation (12) – what is the VSVR?

7) Line 278 “The Fabry-Perot resonant cavity monopole antenna model

Line 335 “the air gap discharge model

Provide reference and describe these models in the corresponding Sections.

And explain – how Chengqiang Liao., L.Z., and Changyue Lu improved the antenna MODEL ?

It is necessary to add References to recent papers in this area and MDPI Journals, for example:

Lv, B.; Zhang, W.; Huang, W.; Li, F.; Li, Y. Narrow Linewidth Half-Open-Cavity Random Laser Assisted by a Three-Grating Ring Resonator for Strain Detection. Sensors 202222, 7882. https://doi.org/10.3390/s22207882.

Varaksin, A.Y.; Ryzhkov, S.V. Turbulence in Two-Phase Flows with Macro-, Micro- and Nanoparticles: A Review. Symmetry 202214, 2433. https://doi.org/10.3390/sym14112433.

Liu, Y.; Isleifson, D.; Shafai, L. Wideband Dual-Polarized Octagonal Cavity-Backed Antenna with Low Cross-Polarization and High Aperture Efficiency. Sensors 202323, 731. https://doi.org/10.3390/s23020731.

Conclusion: Minor revision of the manuscript is required.

Reviewer 3 Report

The authors presented PD wideband high-gain resonant cavity UHF antenna. Although the proposed antenna seems to be new, the reviewer cannot recommend publication of the manuscript in Sensors due to the following issues:

 1. The authors claimed that "It is generally considered that the PD detection bandwidth of power equipment is when the VSWR ≤ 5 [26-31]." However, usually, VSWR ≤ 2 is generally acceptable for good antenna matching performance. In fact, according to [31], it is clearly mentioned that "Given the uncertainty of antenna design, VSWR<2.0 is generally required".

2. The unit of antenna gain is dBi, not dB.

3.  When designing the antenna, the authors use Eq. (4) as the equivalent dielectric constant of the substrate below the monopole antenna. Howeverm Eq. (4) is valid for microstrip transmission lines or microstrip patch antenna (ground exists below the substrate). In fact, for the proposed antenna, the equivalent dielectric constant of the substrate is relatively lower than Eq. (4). Details on this issue can be found in 

A. Abbosh, "Accurate effective permittivity calculation of printed center-fed dipoles and its application to quasi Yagi-Uda antennas," IEEE Transactions on Antennas and Propagation, vol. 61, no. 4, pp. 2297-2300, Apr. 2013.

Some sentences are not clearly written. For example, the following sentence is not clear:

"The area where the power equipment generates PD is not fixed, resulting in the energy spectrum distribution of the electromagnetic wave signal radiated by PD in different 38 types and different insulated environments is not constant."

Round 2

Reviewer 3 Report

The authors have revised the manuscript according to my comments.

It is better to consult your manuscirpt by a native speaker.